# Stimuli-Responsive Track-Etched Membranes for Separation of Water–Oil Emulsions

**DOI:** 10.3390/membranes13050523

**Published:** 2023-05-17

**Authors:** Indira B. Muslimova, Zh K. Zhatkanbayeva, Dias D. Omertasov, Galina B. Melnikova, Arman B. Yeszhanov, Olgun Güven, Sergei A. Chizhik, Maxim V. Zdorovets, Ilya V. Korolkov

**Affiliations:** 1Laboratory of Engineering Profile, L.N. Gumilyov Eurasian National University, Satpaev Str., 5, Astana 010008, Kazakhstan; zhanna01011973@mail.ru (Z.K.Z.); dias2101@mail.ru (D.D.O.); galachkax@gmail.com (G.B.M.); arman_e7@mail.ru (A.B.Y.); mzdorovets@gmail.com (M.V.Z.); 2The Institute of Nuclear Physics, Ibragimov Str., 1, Almaty 050032, Kazakhstan; 3A.V. Luikov Heat and Mass Transfer Institute of the National Academy of Sciences of Belarus, P. Brovki Str., 15, 220072 Minsk, Belarus; chizhik@gmail.ru; 4Department of Chemistry, Hacettepe University, Beytepe, Ankara 06800, Turkey; guven@hacettepe.edu.tr; 5Department of Intelligent Information Technology, Ural Federal University, Mira Str. 19, 620002 Ekaterinburg, Russia

**Keywords:** stimuli-responsive polymers, block copolymerization, track-etched membranes, surface modification, graft polymerization, water–oil emulsions separation

## Abstract

In this work, we have developed a method for the preparation of pH-responsive track-etched membranes (TeMs) based on poly(ethylene terephthalate) (PET) with pore diameters of 2.0 ± 0.1 μm of cylindrical shape by RAFT block copolymerization of styrene (ST) and 4-vinylpyridine (4-VP) to be used in the separation of water–oil emulsions. The influence of the monomer concentration (1–4 vol%), the molar ratio of RAFT agent: initiator (1:2–1:100) and the grafting time (30–120 min) on the contact angle (CA) was studied. The optimal conditions for ST and 4-VP grafting were found. The obtained membranes showed pH-responsive properties: at pH 7–9, the membrane was hydrophobic with a CA of 95°; at pH 2, the CA decreased to 52°, which was due to the protonated grafted layer of poly-4-vinylpyridine (P4VP), which had an isoelectric point of pI = 3.2. The obtained membranes with controlled hydrophobic-hydrophilic properties were tested by separating the direct and reverse “oil–water” emulsions. The stability of the hydrophobic membrane was studied for 8 cycles. The degree of purification was in the range of 95–100%.

## 1. Introduction

Oil pollution of water is an environmental challenge that requires efficient separation methods. One of the most promising technologies in this area is membrane separation [1]. There is a pressing need for effective and efficient methods to separate water–oil emulsions [2]. Membrane filtration is a commonly used technique for the separation of water and oil, but conventional membranes have limitations, such as low selectivity and low efficiency. Nowadays, controllable wettability surfaces are of increasing interest in the field of water–oil emulsion separation. The combination of superhydrophilic/superoleophilic [3,4,5] and superhydrophobic/superoleophilic [6,7] properties increase membrane selectivity in separating water–oil emulsions by allowing one liquid to pass through and repel the other. The authors [4] prepared a superhydrophilic (water CA was 1.4°) and a subsea superoleophobic (subsea oil CA was 163.4°) PVDF membrane modified with graphene oxide, phytic acid and iron (III) for oil-in-water emulsion separation. At the same time, the development of stimuli-responsive polymers has opened new possibilities for the separation of water–oil emulsions. These polymers are capable of undergoing changes in their physical and chemical properties, such as shape, size, or permeability in response to external stimuli, such as pH, temperature, ionic strength, etc. By exploiting these responsive properties, it is possible to achieve highly selective and efficient separation of oil–water emulsions.

There are many articles reporting the use of stimuli-responsive membranes for the separation of water–oil emulsions using mainly pH-sensitive polymers, such as P4VP [8], poly(acrylic acid) [9], and poly(methacrylic acid) [10]. Pyridine, carboxyl, and tertiary amine groups are the typical pH-sensitive functional groups, which are capable of accepting or donating protons when the pH changes. When the pH is below the pI of the ionizable groups on the membrane, the groups are protonated, making the membrane hydrophobic (repels water and attracts non-polar substances). As the pH increases above the pI, the groups become deprotonated, and the membrane becomes more hydrophilic. The transition between these two modes depends on the pH of the surrounding solution. The most popular base materials for incorporating stimuli-responsive polymers are nanofiber and hollow fiber membranes [11,12,13,14], meshes [15], graphene foam [16], and porous anodic aluminum oxide [17]. On the other hand, there are few studies on the use of track-etched membranes (TeMs) in the separation of water–oil emulsions [1], and almost no reports on the use of stimuli-responsive TeMs for this purpose. TeMs prepared from thin polymeric films by irradiation with accelerated heavy ions have unique characteristics, such as narrow pore size distribution and precise control of the number of pores per cm^2^. Therefore, it is possible to use these types of membranes as model membranes and they can be modified to have high selectivity of separation [18,19,20,21].

In this paper, a method is proposed to increase the efficiency of separation of water–oil emulsions by grafting a pH-sensitive block copolymer consisting of hydrophobic polystyrene (PS) units and amphiphilic P4VP units by using reversible addition-fragmentation chain transfer (RAFT) polymerization [22,23]. The general scheme of RAFT polymerization to obtain block polymer and to avoid uncontrolled growth of grafting chains to not block and fill the channels of TeMs, is shown in Figure 1. The RAFT polymerization method involves the use of a RAFT agent to control the polymerization process, leading to the formation of well-defined copolymers with a precisely controlled composition and molecular weight distribution [24,25].

By carefully adjusting the reaction conditions, such as the concentration of monomers and the RAFT agent, as well as the reaction temperature and time, the length and composition of each block can be precisely controlled. The obtained pH-responsive membranes with controlled hydrophobic-hydrophilic properties were tested by separating the direct and reverse “oil–water” emulsions. The stability of the hydrophobic membrane was checked for 8 cycles. The degree of purification was in the range of 95–100%.

## 2. Materials and Methods

### 2.1. Chemicals

Styrene (ST), 4-vinylpyridine (4-VP), 2-(dodecylthiocarbonothioylthio)propionic acid and isopropyl alcohol (IPA) were supplied by Sigma Aldrich. All other chemicals and solvents, such as chloroform, cetane, hexane, cyclohexane, benzophenone (BP), acetic acid, muriatic acid, N,N-dimethylformamide (DMF), and sodium hydroxide had purity of analytical grade.

### 2.2. Preparation and Modification of Track-Etched Membranes

PET films of the Hostaphan^®^ trademark manufactured by Mitsubishi Polyester Film (Germany) with a nominal thickness of 23 µm were irradiated with Kr ions with energy of 1.75 MeV/nucleon at the cyclotron DC-60 (Astana, Kazakhstan) with a pore density of 1 × 10^6^ ions/cm^2^ in order to form latent tracks. Photooxidation of radiolysis products was performed by photosensitization of PET TeMs under a UV lamp with a wavelength of 254 nm and radiation power of 12 W at a distance of 10 cm for 30 min on each side. Photosensitization increases the etching rate of ion-tracks [26]. During the etching step, the polymer backbones are destroyed with the formation of -OH and -COOH groups at the chain terminals [27,28,29], which allows the formation of PET TeMs pores with diameters of 2.0 ± 0.1 µm with cylindrical shape.

Schematically, the process of membrane preparation and modification is presented in Figure 2. Benzophenone (BP) was used with DMF to remove excess BP, dried, weighed, and placed in a vessel containing the reaction mixture. Before polymerization, the reaction media was bubbled with argon.

Modification of PET TeMs was performed on both sides of the membrane by RAFT graft block copolymerization in two consecutive steps: ST grafting and 4-VP grafting. IPA was used as the reaction medium because of its good solubility, availability, and transparency in the UV range.

Irradiation was performed under an OSRAM Ultra Vitalux E27 UV lamp with the following characteristics: UVA—315–400 nm, 13.6 W; UVB—280–315 nm, 3.0 V. The container with the reaction mixture and the initial sample was hermetically sealed with a film made of PVC.

### 2.3. Methods of Characterization

FTIR spectra were recorded on an FTIR InfraLUM FT-08 spectrometer to study the functional groups before and after modification. Measurements were taken in the range of 400–4000 cm^−1^. Spectra in 20 scans at 2 cm^−1^ resolution using an ATR (Pike) attachment were recorded at room temperature. Spectral analysis was performed using the SpectraLUM^®^ program.

AFM was carried out on NT-206 devices ALC “Microtestmachines” to study the morphology and local mechanical properties (adhesion force and elasticity modulus) of the surfaces of micro- and nanometer-sized features. The array of obtained data was processed using the specialized software Surface Explorer. Based on the results of AFM scanning of 10 × 10 µm^2^ areas with a resolution of 128 × 128 points, the average (R_a_, nm) and root mean square (R_q_, nm) roughness of the sample surface were determined for 5 different scanned areas. The elasticity modulus (E, MPa) and adhesion force (F, nN) were calculated on the basis of indentation curves using the Johnson–Kendall–Roberts model [30].

The water contact angle (CA) characterizes the permeability and hydrophilic-hydrophobic properties of the membrane. The CA was determined on a Kruss DSA 100E device using the lying drop method based on the drop height and interfacial boundary values. To calculate the values of the surface free energy (ω, mN/m) and its specific polar component (γ^p^, mN/m) according to the OWRK method (Owens, Wendt, Rabel & Kelble), distilled water and diiodomethane were used as test liquids. The accuracy of the CA measurement was ±0.10°. The change in the hydrophilic-hydrophobic properties of the membranes was monitored by measuring the CA at different pH values of 2, 7 and 9.

SEM-EDX analysis was performed using an Hitachi TM 3030 with a Bruker XFlash MIN SVE microanalysis system at 15 kV to study the surface of the TeMs both before and after grafting. The pore diameters and their morphology were also monitored using SemImage software.

DLS was performed on an Analysette 22 MicroTec plus using a wet dispersion unit.

The degree of grafting was determined according to Equation (1) by the difference in membrane weight before and after the modification as shown in previously published works [1,31].
(1)$$\eta =\frac{\left(m_{2}-m_{1}\right)}{m_{1}}\times 100\%$$
where η—is the degree of grafting, $m_{1}$—is the weight of the membrane before grafting, and $m_{2}$—is the weight of the membrane after grafting. To remove electrostatic charge, antistatic ionizer was used before weighing.

A Specord-250 spectrophotometer Analytik Jena was used to study the reaction mixture before and after irradiation at the wavelength range of 190–800 nm. Monomer conversion and the theoretical molecular weight of the polymer were calculated according to Equations (2) and (3).
(2)$$Convn=\left(1-\frac{C_{m}^{\prime }}{C_{m}}\right)\times 100\%$$
(3)Mwtheo=CmCRAFT×Mwm×Convn100%+MwRAFT
where Convn—is the percent conversion, $C_{m}^{\prime }$—is the concentration of the monomer remaining after polymerization, $C_{m}$—is the initial concentration of the monomer, ${Mw}_{theo}$—is the theoretical molecular weight of the polymer, $C_{RAFT}$—is the concentration of the RAFT agent, ${Mw}_{m}$—is the molecular weight of the monomer, and ${Mw}_{RAFT}$—is the molecular weight of the RAFT agent.

### 2.4. Testing of pH-Responsive PET TeMs in the Separation of Water–Oil Emulsions

Membranes were tested on the setup shown in our previous paper [1]. The filtration unit was connected to a vacuum pump with a vacuum controller. Filtration was performed at a vacuum of 900–950 mbar. Cetane, hexane, and cyclohexane were used as the organic part. The water–oil emulsion model was prepared using an IKA T18 digital Ultra-Turrax disperser. The ratio for the reverse emulsion oil:water (pH9) = 100:1, 20:1 (vol%) and for the direct emulsion oil:water (pH2) = 1:100 (vol%). The droplet size (D, µm) of the emulsion before and after separation was controlled by DLS analysis in the range 0.1–45 µm. The membrane was pre-soaked for 30 min in water at pH 9 to separate the reverse emulsion, and in pH 2 to separate the direct emulsion. After filtering the direct emulsion through the membrane, the oil was removed, and the purified water was collected in a beaker. The flux of the filtered water–oil mixture was calculated from Equation (4) as shown in our previously published works [1,31].
(4)F=V(S×t)
where F—is the flux, L/h·m^2^; V—is the volume of the water that permeates through the membrane, L; S—is the filtration area of PET TeMs, m^2^; and t—is the flow time, h.

The volume of oil collected after separation was measured and the separation efficiency was calculated using the following Equation (5):(5)$$R=\frac{V_{2}}{V_{1}}\times 100\%$$
where R—is the separation efficiency, $V_{1}$—is the volume of oil in the oil-in-water emulsion before separation; and $V_{2}$—is the volume of oil collected after separation.

Similarly, for the reverse emulsion, the volume of oil passing through the membrane and the volume of water remaining after separation were used to calculate the efficiency.

## 3. Results and Discussion

The introduction of functional groups on the walls of pores of PET TeMs is a challenging task comprised of consecutive grafting of hydrophobic (PS) and hydrophilic (P4VP) polymers in a controlled manner so as not to clog the pores by polymer chains growing radially from the inner surface of the pores as well as further modification as a function of pH. In order to achieve grafted block copolymers (PS-b-P4VP) with controlled chain lengths, reversible addition-fragmentation chain-transfer (RAFT) polymerization was applied in pre-irradiation mode. P4VP with an isoelectric point of 3.2 was selected as the pH-responsive component of the copolymer as it acquires positive charge at pH values less than 3.2 exhibiting hydrophilic properties.

### 3.1. Graft Polymerization of Styrene

The preparation of a stimulus-responsive membrane surface was performed in two steps. In the first step, hydrophobic PET TeMs were obtained by UV-initiated grafting by RAFT polymerization of ST. The choice of ST as a hydrophobic agent was due to the fact that it is one of the most affordable, cheap, and widely used hydrophobic monomers. The influence of the monomer concentration (c = 1–4 vol.%), molar ratio of RAFT agent: initiator (1:2–1:100), and the grafting time (t = 30–120 min) on the degree of grafting (%), monomer conversion (%), and theoretical molecular weight of polymer (Mw, g/mol) was studied. The obtained data are summarized in Table 1.

Appendix A shows the dependence of the time of UV irradiation (30, 60 and 120 min), the molar ratio of RAFT agent:initiator (1:2, 1:10 and 1:100) and the monomer concentration (1, 2 and 4 vol.%) on the grafting degree and conversion of the monomer (the distance from the UV lamp was constant at 7.5 cm). Isopropyl alcohol was used as a solvent, since it is transparent in the UV-visible region and it can dissolve all reagents used in this work.

With increasing time of UV irradiation (Appendix A) and ST concentration (Appendix A), there was an increase in the degree of grafting (from 0.31 to 12% and from 0.21 to 8.6%, respectively) and monomer conversion (from 13 to 42% and from 17 to 36%, respectively). As can be seen from Appendix A, the degree of grafting increased slightly with an increase in the molar ratio RAFT agent: initiator from 1:2 to 1:10 (to 2.6%) and increased at 1:100 (3.2%). The ST conversion values increased linearly with an increase in the molar ratio RAFT agent: initiator from 1:2 to 1:100 (from 9.6 to 53%) at constant time (60 min) and concentration of ST (2 vol.%). The molar ratio of the RAFT agent to benzophenone had a great influence on the conversion. With an increase in the amount of initiator in the mixture, the polymerization rate increased. Based on this, the maximum value of ST conversion was found to reach 53% (Appendix A), with a molar ratio of 1:100, monomer concentration—2 vol.% and time of UV irradiation—60 min. This trend was in accordance with [32] where the effect of the amount of ascorbic acid initiator on the RAFT copolymerization of styrene and maleic anhydride was studied. It was found that increasing the equivalent of the initiator to the RAFT agent led to an increase in monomer conversion from 17 to 70%. The theoretical molecular weight of PS increased with increasing time of UV irradiation, the molar ratio of the RAFT agent to benzophenone, and the ST concentration.

The possibility of decomposition of the RAFT agent was experimentally studied. On the UV-vis spectra of the RAFT agent, there was a small shift after an hour of irradiation under the conditions of graft polymerization; however, the intensity of the peak was not found to change. Thus, the concentration of the RAFT agent was not affected during UV-irradiation.

The hydrophobic properties of the surface of the PS modified PET TeMs were estimated by the values of the contact angle. The results of the CA measurements for each of the PET TeMs samples before and after the graft polymerization of the PS are shown in Table 1. The CA of the modified samples reached a maximum value of 97°. Modification with a layer of PS increased the CA of the PET TeMs from 67° to 97°. The highest value of CA 97° was achieved at a PS concentration of 2 vol.%, an irradiation time of 60 min, and a molar ratio of RAFT agent:initiator = 1:10. Reducing the molar ratio of the RAFT agent and initiator to 1:2 lowered the CA to 88°, which corresponded to the lowest conversion (9.61%) under these conditions (Table 1). This was due to the lower concentration of the benzophenone initiator in the reaction mixture. The membrane surface became hydrophobic (CA was more than 90°) regardless of the change in the concentration of the monomer and the irradiation time.

The results of the morphology and local mechanical properties of the pristine and modified membranes are summarized in Table 2.

As is well known, polystyrene displays typical hydrophobic behavior with water CA of 93 ± 1.0° [33]. The contact angle of PS grafted on PET obtained in this work (92–97°, Table 1) showed that the PET surface was fully covered by the PS graft chains.

As can be seen from the data in Table 2, the values of free energy and its polar part were in agreement with the CA values presented in Table 1. The non-polar PS (-H_2_C-CH-C_6_H_5_-)n covalently bound with the PET surface and covered the polar carbonyl/carboxyl-C=O (COOH) and hydroxyl -OH groups of PET, forming a more hydrophobic layer on the membrane surface. With increasing concentration of PS, irradiation time and the ratio of the RAFT agent and initiator, the polar part of the free surface energy decreased, so the membrane surface became more hydrophobic due to the increasing degree of grafting of PS, as shown in Figure 1. The samples with the highest CA value of 97° had the lowest free surface energy of 42 mN/m and its polar part of 0.01 mN/m. The pore diameter decreased significantly with grafting ST on the membrane pore surface. An increase in the grafting time reduced the diameter of the pores from 2.0 pristine PET TeMs to 1.7 μm after styrene modification for 120 min (Table 1).

In addition to the surface energy, CA also depends on the surface roughness (average R_a_ and R_q_ indices). The data in Table 2 show that the roughness and pore diameter values did not change significantly with increasing RAFT agent:initiator ratios. The increase in grafting time from 30 to 120 min allowed the formation of a dense and regular PS layer on the membrane surface (Ra = 2.3 nm and Rq = 2.9 nm).

The 10 × 10 µm^2^ AFM images shown in Figure 3, with a resolution of 128 × 128 dots across the five scanning areas, were obtained using tapping mode.

Grafting PS in low concentration (1 vol.%) increased the values of the arithmetical mean (Ra) and the roughness mean (Rq) from 12 nm and 16 nm to 14 nm and 20 nm, respectively, whereas the pore diameter decreased from 2.0 to 1.8 µm, as shown in Table 1. Performing styrene grafting at 2 vol.% (grafting degree: 2.6 ± 0.02%) (Figure 3c) preserved the microscale structure of the pristine sample, slightly decreasing Ra = 10 and Rq = 15 nm and the pore diameter to 1.8 µm. A further increase in styrene concentration (Figure 3d) to 4 vol.% (grafting degree: 8.65 ± 1.03%) flattened the surface structure to form a dense layer (Ra = 2.2 nm and Rq = 2.8 nm).

The EDX mapping presented in Figure 4 shows the uniform distribution of elements on the membrane surface. The data collected from the EDX analysis are summarized in Table 3.

Figure 4 and Table 3 show that, after PS grafting, sulfur was present on the membrane surface in the very small amount of 0.06%, the amount of carbon increased from 70 to 71%, and oxygen decreased from 30 to 29%.

SEM images of the surface of PET TeMs depending on grafting time are presented in Figure 5. As can be seen from this figure, there was a consistent decrease in the pore diameter with increasing time of grafting to 30, 60 and 120 min (corresponding to 0.31 ± 0.24, 2.6 ± 0.02 and 12 ± 1.1% grafting degree) from 2.0 μm to 1.9, 1.8 and 1.7 μm, respectively (Table 1), while the morphological structure of the surface became smoother in microscale. The decrease in pore size was due to the formation of a PS graft layer.

FTIR-ATR spectroscopy showed the chemical changes on PET TeMs surface after grafting styrene. A typical FTIR-ATR spectrum of pristine PET TeMs (Figure 6) consisted of main absorption peaks at 2972 cm^−1^ (aromatic C-H), 2910 cm^−1^ (aliphatic C-H), 1715 cm^−1^ (C=O), 1471 cm^−1^ (CH_2_ bending), 1410 cm^−1^ (ring CH in plane bending), 1341 cm^−1^ (CH_2_ stretching), 1238 cm^−1^ (C(=O)-O stretching), 1018 cm^−1^ (ring CCC bending), 970 cm^−1^ (O-CH_2_ stretching), and 847 cm^−1^ (ring CC stretching). The wavenumbers and assignments of the spectra were in good agreement with previously published works [1,31].

Graft polymerization of PS led to the appearance of new peaks characteristic of PS: 1580, 1450 cm^−1^ (CH_2_-deformation), 700 cm^−1^ (CH_2_-rocking mode), 530 cm^−1^ (CH-benzene ring), and 1480 cm^−1^ (C = C-benzene ring). The most characteristic changes in the spectrum after grafting as a function of grafting time were observed for the peak at 700 cm^−1^ (Figure 6b). For a quantitative assessment, the absorbance values of A_700_/A_1410_ were calculated from the corresponding peak areas, and the results are presented in Table 4. As the ST concentration, grafting time, and molar ratio of the RAFT agent and initiator increased, the spectroscopic indices of Area_700_/Aarea_1410_ increased from 0.30 to 28.

Thus, as a result of PS grafting, a hydrophobic layer was formed on the membrane surface, and, according to the above results, the optimal conditions for PET TeMs modification by UV-initiated RAFT graft polymerization of ST, leading to maximum hydrophobization of membranes with preservation of pore structure, were:−grafting time—60 min,−molar ratio RAFT agent:initiator = 1:10,−monomer concentration—2 vol.%.

### 3.2. Graft Copolymerization of 4-Vinylpyridine on PET TeMs-g-PS

In the second part of this work, 4-vinylpyridine (4-VP) was grafted from the chain ends of PS already grafted on PET-TeMs by RAFT polymerization to obtain P4VP-b-PS block copolymers inside the channels of the tracks. The influence of the monomer concentration (c = 1–4 vol.%), molar ratio of RAFT agent: initiator (1:2–1:100) and the grafting time (t = 30–120 min) on the degree of grafting (%), monomer conversion (%), and the theoretical molecular weight of polymer (Mw, g/mol) were also studied at this stage of modification (the distance from the UV-source was kept constant at 7.5 cm). IPA was also used as the solvent and 2-(dodecylthiocarbonothioylthio) propionic acid as the RAFT agent. The data obtained are summarized in Table 5.

Appendix A shows the effect of the UV irradiation time (30,60 and 120 min), the molar ratio of RAFT agent:initiator (1:2, 1:10 and 1:100), and the 4-VP concentration (1, 2 and 4 vol.%) on the grafting degree and conversion of the monomer.

As can be seen in Appendix A, the degree of grafting of 4-VP increased with decrease in the molar ratio of the RAFT agent to the initiator from 1:2 to 1:100 and increase in the concentration of the monomer from 1 to 4 Vol.% (from 0.68 to 1.4% and from 0.67 to 3.3%, respectively). As shown in Appendix A, an increase in the time of UV irradiation (30, 45 and 60 min) of 4-VP did not affect the conversion (9.5, 9.6 and 9.03%, respectively). The degree of grafting increased from 0.5 to 1.2% with increase in the time of UV irradiation from 30 to 45 min; subsequent increase in time to 60 min reduced the degree of grafting to 0.65% as a result of possible polymer destruction (Appendix A). This was also indicated by the fact that, in the FTIR spectra, which will be presented later in Figure 10, we observed an increase in the peak related to P4VP at 1599 cm^−1^ (pyridine ring band), indicating that an increase in the content of the grafted P4VP had occurred, and that the decrease in weight was probably due to weak ultraviolet (UV) resistance [34]. At the same time, the theoretical molecular weight of P4VP increased with increasing molar ratio of the RAFT agent to the benzophenone and 4-VP concentration. However, by increasing the grafting time from 45 to 60 min, a slight decrease in theoretical molecular weight from 220 ± 9.7 × 10^3^ to 197 ± 12 × 10^3^ g/mol occurred.

Increase in the monomer concentration from 1 to 2 vol.% increased the conversion from 0.2 to 9.6%; however, a subsequent increase in concentration to 4% led to a decrease in conversion to 5.6% despite an increase in the degree of grafting. The grafting degree also depended on the molar ratio of the RAFT agent to the initiator: with change in the ratio from 1:2 to 1:100, the conversion of the monomer increased from 5.8 to 13%, as shown in Appendix A.

The effect of pH on the hydrophilic/hydrophobic properties of the obtained membranes was studied (Table 5). From the data presented in Table 5, the following tendency was observed: the greater the degree of grafting of 4-VP on the surface of the PET TeMs-g-PS, the lower the CA of the obtained samples at pH 7. That is, in the grafted samples with a monomer concentration of 4 vol.% and ratio of RAFT agent:initiator = 1:100, which corresponded to the degrees of grafting of 3.3 and 1.4%, respectively, the main contribution to CA was provided by the grafted P4VP layer. If the grafting degree of 4-VP was high enough, the samples were hydrophilic (<90°) in the pH range from 2 to 9. However, if the degree of grafting was not high enough, then the samples remained hydrophobic.

pH-responsive PET TeMs were obtained under the following conditions: 4-VP concentration of 2 vol.%, molar ratio RAFT agent:initiator = 1:10, and time—45 min. Under these conditions the largest difference in CA was observed (Figure 7): at pH 2 CA was 58°, at pH 9 CA was 95°. Furthermore, it should be noted that the obtained pH-responsive PET TeMs, after soaking at pH9 (membrane became hydrophobic), instantly allowed pure cetane, hexane, and cyclohexane to pass through the pores of the membranes. However, after soaking the membrane at pH2, the CA of cetane, hexane, and cyclohexane was 29, 28, and 26°, respectively, and during filtration testing, the oil remained on the surface of the membrane and did not leak out.

Figure 7d shows that the obtained sample after P4VP grafting (degree of grafting 1.2%) did not change the surface CA, that is, the main contribution to the surface CA in a neutral medium was provided by the grafted PS. Thus, we can conclude that pH-responsive membranes were obtained whose surface at pH > 3.2 was hydrophobic because the P4VP grafted chains were in a twisted non-ionized state, while in an acidic environment the membrane surface was hydrophilic because, at pH < 3.2, the P4VP grafted chains acquired a positive charge and became extended, as schematically shown in Figure 2.

Table 6 presents the results of the study of the morphology and local mechanical properties (adhesion force, elastic modulus, roughness, surface energy and its polar part) of the surfaces of the obtained membranes.

As can be seen from the data in Table 6, the values of the free energy and its polar part agreed with the CA values presented in Table 5. The polar part of the free energy after grafting of hydrophobic PS decreased from 6.7 to 0.01 mN/m and, after grafting of hydrophilic 4-VP, increased again to 2.01 mN/m at a monomer concentration of 4 vol.%, which corresponded to 3.3% degree of grafting. At small monomer concentrations of 1 vol.% and irradiation times of 30 min and 60 min, the polar part of the free energy remained unchanged due to the small degree of grafting. P4VP grafting led to an insignificant increase in the adhesion force of the modified samples up to 28 nN; the value of the elastic modulus did not change on average. The thickness of the grafted layers of PS/P4VP was about 40 nm according to AFM.

Based on the AFM studies, the images shown in Figure 8, with a size of 10 × 10 μm^2^ and a resolution of 128 × 128 points in five scanning areas, were obtained. With increase in the monomer concentration up to 4 vol.%, the surface roughness of the membranes modified with 4-VP under optimal conditions increased up to Ra = 16 nm and Rq = 22 nm, while increasing the grafting time allowed the formation of a layer with Ra = 22 nm and Rq = 29 nm.

As can be seen from Figure 8, with increasing concentration of 4-VP, the surface morphology in microscale became smoother and the pore diameter slightly decreased with increasing 4-VP concentration.

The EDX mapping (Figure 9) shows the uniform distribution of C, O, N, S elements on the surface of the 4-VP grafted surface at concentration 2 vol.%, and molar ratio RAFT agent:initiator = 1:10 for 45 min grafted membranes on PET TeMs-g-PS. The data obtained from the EDX analysis are summarized in Table 7.

Table 7 and Figure 9 show that 5.9% nitrogen was present on the surface of the modified membrane, which is a direct proof of P4VP grafting. The amount of carbon after P4VP grafting decreased to 67%, and of oxygen to 27%. Sulfur was also present in small amounts (0.05%). P4VP grafting onto the PET TeMs-g-PS samples reduced the membrane pore diameter to 1.7 µm (Table 5). The cylindrical pore geometry was retained. The increase in 4-VP concentration from 1 to 4 vol.%, grafting time from 30 to 60 min, and decrease in molar ratio RAFT agent:benzophenone from 1:2 to 1:100 did not significantly affect the pore diameter (1.7 µm).

FTIR-ATR spectroscopy showed the chemical surface changes of PET TeMs-g-PS after 4-VP grafting (Figure 10). The FTIR-ATR spectra of the original PET TeMs and PET TeMs-g-PS consisted of the main absorption peaks, as shown in Figure 6.

Graft polymerization of 4-VP led to the appearance of a new peak characteristic of P4VP at 1599 cm^−1^ (C = C aryl., pyridine ring). For a quantitative assessment of the graft polymerization of 4-VP, the values of the Area_1599_/Area_1410_ spectroscopic indexes were calculated from the respective peak areas. As the grafting time increased from 30 to 45 and 60 min, a consistent increase in the indices of Area_1599_/Area_1410_ from 0.03 to 0.06 and 0.10, respectively, were observed.

Thus, the UV-initiated RAFT grafting of P4VP onto the surface of membranes modified with hydrophobic PS under optimal grafting conditions resulted in pH-responsive PET TeMs, which were hydrophobic at pH 7 (97°) and pH 9 (95°) and hydrophilic at pH 2 (58°). According to the above results, the optimum conditions for the modification of PET TeMs by UV-initiated RAFT grafting of PS-b-P4VP block copolymers inside the channels of PET TeMs, to achieve controlled hydrophilic/hydrophobic membrane properties with retained pore structure, were:−grafting time: 45 min,−molar ratio RAFT agent:initiator = 1:10−monomer concentration: 2 vol.%.

### 3.3. Testing of pH-Responsive Membranes in Water–Oil Emulsion Separation

The obtained pH-responsive membranes were tested in the separation of a water–oil emulsion under the two cases of direct emulsion, water(pH2):oil = 100:1 and reverse emulsion, water(pH9):oil = 1:100 and 1:20, at a constant vacuum pressure of 900–950 mbar. Cetane, hexane, and cyclohexane were used as the organic part. The flux of the filtered oil–water mixture is presented in Figure 11. The separation efficiency was in the range of 95–100%. Pristine PET TeMs were tested in the same way; however, since pristine PET TeMs have semi-hydrophobic properties, it was observed that both the water and organic parts passed through the pores of the membranes—separation was not observed.

The use of pH-responsive PET TeMs allowed the separation of both direct and reverse emulsion with high separation efficiency (R) (95–100%). DLS analysis showed that the droplet size (D, μm) for the direct emulsions was 0.6, 3.1, and 4.2 μm for cetane, hexane, and cyclohexane as the organic part, respectively. The droplet size of the reverse emulsions was 4.9 μm. After separation, DLS analysis did not detect an emulsion even after dispergation of the filtrate using a IKA T18 digital Ultra-Turrax disperser. The results are summarized in Table 8.

The stability of the hydrophobic membrane was studied over eight cycles. When separating a reverse emulsion water:cetane, the flux of separation with increasing separation cycles varied within a small range between 560 L/h·m^2^ and 480 L/h·m^2^ L/h·m^2^, which confirmed the stability of the membrane, but was considerably lower compared to the flux of separation in direct emulsion separation (maximum flux value 5200 L/h·m^2^), probably due to the higher viscosity of cetane compared to water. The gradual decrease in fluxes of the direct water:cetane emulsion from 5200 to 2500 L/h·m^2^ was probably due to membrane pore contamination (pore diameter 1.7 μm) with the relatively long-chain cetane. Nevertheless, despite the membrane contamination, the separation efficiency in each cycle was high (97–100%). The fluxes in the direct and reverse hexane:water emulsions also decreased with increasing separation cycles and ranged from 1400 to 860 L/h·m^2^ and from 7400 to 5900 L/h·m^2^, respectively. The separation efficiency was also high (95–100%). When separating the water:cyclohexane emulsions, the vacuum pressure was raised to 950 mbar, because, when the vacuum was set below 950 mbar, cyclohexane seeped through the pores of the membranes along with water and there was no separation of the emulsion. With increasing droplet size (Table 8) of the direct emulsions water:cetane (0.6 μm), water:hexane (3.1 μm) and water:cyclohexane (4.5 μm), the coalescence of drops was accelerated [35], and the average flux values decreased to 3600 ± 870, 1100 ± 170 and 670 ± 110 L/h·m^2^, respectively, which was most probably related to the formation of a film on the membrane surface of the organic part.

The PET TeMs membranes modified by UV-initiated RAFT-block copolymerization of PS and P4VP acquired controlled hydrophilic-hydrophobic properties which could be adjusted by changing the pH of the medium. These membranes were successfully tested in the separation of direct as well as reverse water–oil emulsions.

The obtained results on the flux and separation efficiency were compared with previously published articles on the separation of water–oil emulsions. The results are presented in Table 9. The prepared pH-responsive PET TeMs have the potential to be used for oil–water separation. The highest fluxes were observed when separating the direct water: cetane emulsions (5200 L/h·m^2^) and the reverse cyclohexane:water emulsions (7400 L/h·m^2^), which were comparable with other membranes and, in some cases, superior to the fluxes of previously developed membranes.

## 4. Conclusions

This study presents a method of preparation of pH-responsive TeMs based on PET by RAFT block copolymerization of ST and 4-VP to be used for the separation of water–oil emulsions. The optimal conditions found for ST grafting were: monomer concentration—2 vol.%, molar ratio RAFT agent:initiator = 1:10, and grafting time 60 min, and for 4-VP grafting: monomer concentration—2 vol.%, molar ratio RAFT agent:initiator = 1:10 and grafting time 60 min. Under these conditions the obtained membranes showed pH-stimuli-responsive properties: in neutral and alkaline media the membrane was hydrophobic with a CA of 95° and membranes had a CA of 52° at pH values below the isoelectric point of P4VP (pI = 3.2). The SEM-EDX results showed that the grafted copolymer was evenly distributed on the membrane surface, the pore diameter of the resulting membranes decreased from 2.0 to 1.7 µm and the pore structure was retained. The obtained membranes with controlled hydrophobic-hydrophilic properties were tested by separating the direct (100:1) and reverse (1:100) “oil–water” emulsions, by using cetane, hexane, and cyclohexane as the organic part. The maximum flux value of 5200 L/h·m^2^ was achieved for the direct cetane-in-water emulsion and 7400 L/min*m^2^ for the reverse water-in-hexane emulsion. The stability of the hydrophobic membrane was studied for eight cycles. The degree of purification was in the range of 95–100%.

## Figures and Tables

**Figure 1 membranes-13-00523-f001:**
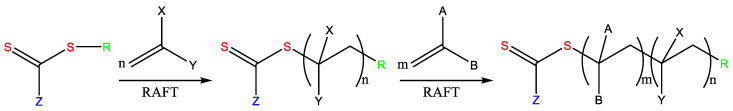
Reaction scheme of block copolymer formation by sequential RAFT polymerization.

**Figure 2 membranes-13-00523-f002:**
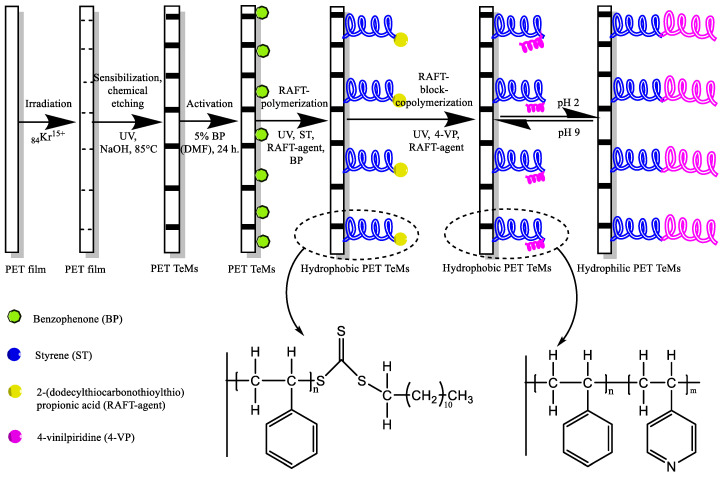
The reaction pathways for obtaining pH-sensitive PET TeMs.

**Figure 3 membranes-13-00523-f003:**
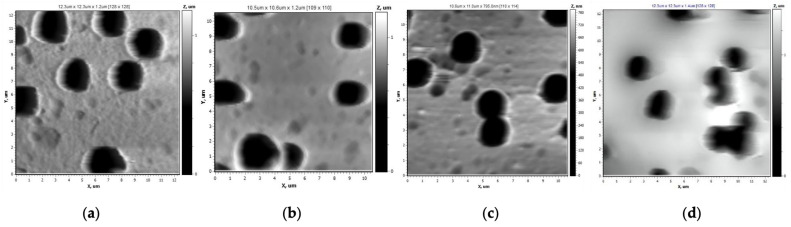
AFM images of the pristine PET TeMs (**a**) and those grafted with PS with monomer concentration of 1, 2 and 4 vol.% ((**b**–**d**), respectively) at 60 min irradiation time and RAFT agent:initiator molar ratio = 1:10 with size 10 × 10 µm^2^.

**Figure 4 membranes-13-00523-f004:**

EDX mapping of elements of the surface of PET TeMs-g-PS (degree of grafting is 2.6 ± 0.02%).

**Figure 5 membranes-13-00523-f005:**
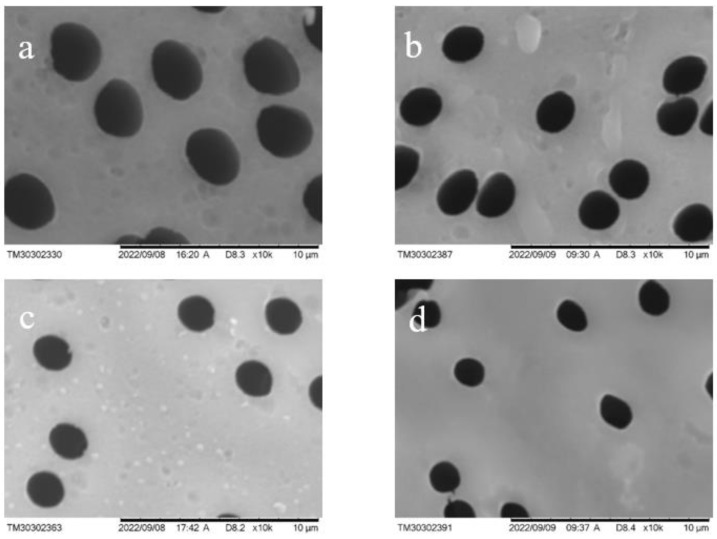
SEM images of PET TeMs surface before (**a**) and after 30 min (**b**), 60 min (**c**) and 120 min (**d**) grafting of styrene (monomer concentration 2 vol.% and molar ratio RAFT agent:initiator = 1:10) (Degree of grafting: 0, 0.31 ± 0.24, 2.6 ± 0.02 and 12 ± 1.1%).

**Figure 6 membranes-13-00523-f006:**
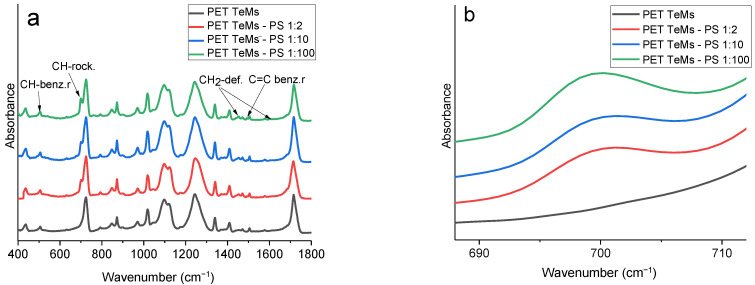
FTIR spectra of the pristine and modified PET TeMs as a function of RAFT agent:initiator molar ratio at constant styrene concentration of 2 vol.% and 60 min irradiation time in the range of 400–1800 cm^−1^ (**a**) and 680–710 cm^−1^ (**b**).

**Figure 7 membranes-13-00523-f007:**
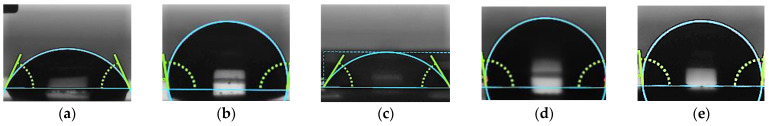
CA of pristine PET TeMs (**a**), PET TeMs-g-PS (**b**) and PET TeMs-g-PS-g-P4VP obtained by graft polymerization of 4-VP (concentration 2 vol.%, molar ratio RAFT agent:initiator = 1:10 for 45 min) at different pH 2 (**c**), pH 7 (**d**) and pH 9 (**e**). (**a**) 67°. (**b**) 97°. (**c**) 58°. (**d**) 97°. (**e**) 95°.

**Figure 8 membranes-13-00523-f008:**
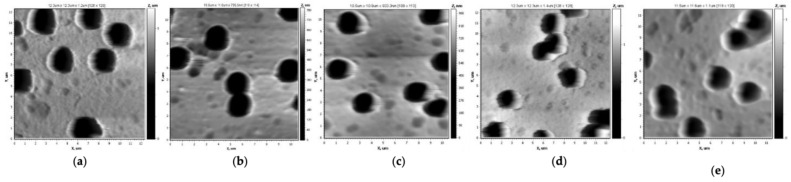
AFM images of pristine PET TeMs (**a**), PET TeMs-g-PS (**b**) and PET TeMs-g-PS-g-P4VP at grafting degree of 0.67 ± 0.03 (**c**), 1.2 ± 0.02 (**d**) and 3.3 ± 0.14% (**e**) obtained at 1, 2 and 4 vol.% monomer concentration, respectively, and constant irradiation time of 45 min with molar ratio of RAFT agent:benzophenone = 1:10.

**Figure 9 membranes-13-00523-f009:**
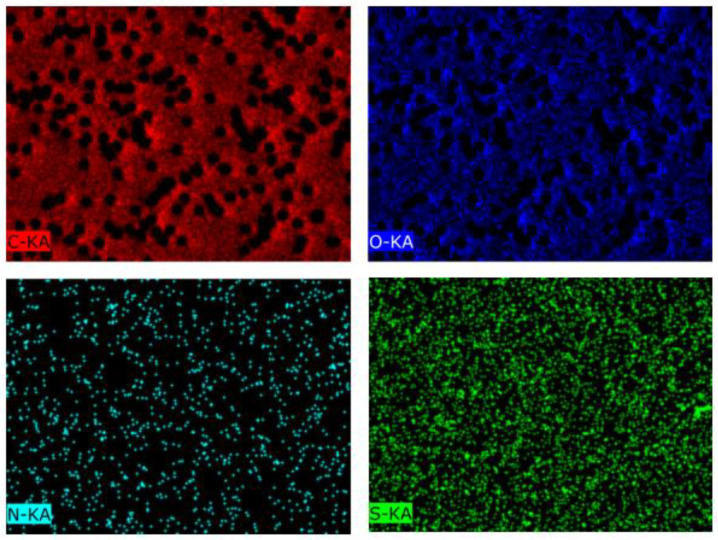
EDX mapping of elements at the surface of PET TeMs-g-PS-g-P4VP (degree of grafting is 1.2 ± 0.02%).

**Figure 10 membranes-13-00523-f010:**
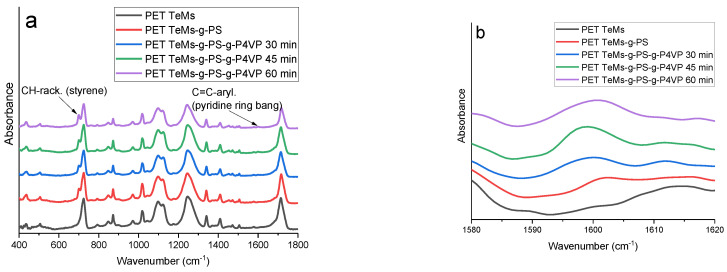
FTIR spectra in the range of 400–1800 cm^−1^ (**a**) and 680–710 cm^−1^ (**b**) of the pristine PET TeMs, PET TeMs-g-PS-g-P4VP obtained at different irradiation time (30, 45 and 60 min) and at constant 4-VP concentration of 2 vol.%, molar ratio of RAFT agent:benzophenone = 1:10.

**Figure 11 membranes-13-00523-f011:**
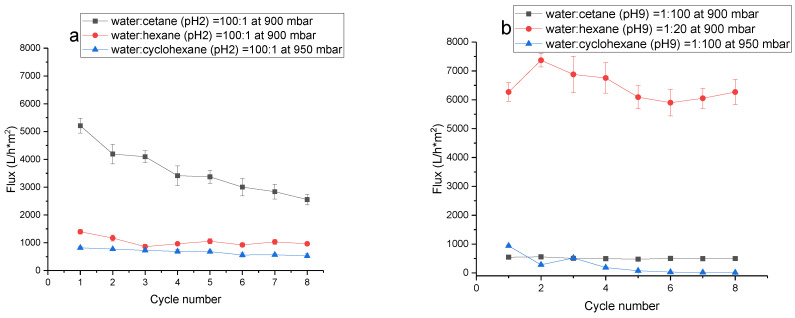
The fluxes of pH-responsive PET TeMs for direct (**a**) and reverse (**b**) emulsions at 900-950 mbar, consisting of organic part and water after each cycle.

**Table 1 membranes-13-00523-t001:** Main parameters of PET TeMs before and after grafting with ST.

Reaction Parameters	Grafting Degree, %	Conversion, %	Theoretical Mw of Polymer, ×10^3^ g/mol	Contact Angle, ±3 °	Pore Diameter, ±0.1 μm
Pristine PET TeMs	-	-	-	67	2.0
Concentration of monomer, vol.% *					
1	0.21 ± 0.101	17 ± 0.30	190 ± 3.3	96	1.8
2	2.6 ± 0.02	25 ± 2.7	590 ± 61	97	1.8
4	8.7 ± 1.03	36 ± 0.89	1600 ± 40	95	1.7
RAFT agent: Initiator, molar ratio **					
1:2	2.6 ± 0.01	9.6 ± 1.2	220 ± 26	88	1.8
1:10	2.6 ± 0.02	25 ± 2.7	590 ± 61	97	1.8
1:100	3.2 ± 0.14	53 ± 8.3	1200 ± 19	94	1.9
Time, min ***					
30	0.31 ± 0.24	13 ± 1.3	290 ± 29	92	1.9
60	2.6 ± 0.02	25 ± 2.7	590 ± 61	97	1.8
120	12 ± 1.1	42 ± 1.3	950 ± 29	96	1.7

*—at constant molar ratio of RAFT agent:Initiator = 1:10 and irradiation time of 60 min. **—at constant time 60 min and concentration of ST 2 vol.%. ***—at constant concentration of ST 2 vol.% and molar ratio of RAFT agent:initiator = 1:10.

**Table 2 membranes-13-00523-t002:** Surface properties of PET TeMs-g-PS under different graft polymerization conditions.

Reaction Parameters	ω, ±0.01 mN/m	γ^p^, ±0.01 mN/m	R_a_, nm	R_q_, nm	F, nN	E, ±200 MPa
Pristine PET TeMs	53	6.7	12 ± 2.5	16 ± 4.5	21 ± 2.5	530
Concentration of monomer, vol.% *						
1	50	0	14 ± 3.1	20 ± 5.0	17 ± 2.8	450
2	42	0.01	10 ± 1.5	15 ± 2.6	32 ± 2.8	570
4	43	0.14	2.2 ± 0.4	2.8 ± 0.5	31 ± 9.7	570
RAFT agent: Initiator, molar ratio **						
1:2	50	0.45	12 ± 3.1	17 ± 5.1	21 ± 2.2	570
1:10	42	0.01	10 ± 1.6	15 ± 2.6	32 ± 2.8	570
1:100	50	0	10 ± 2.2	14 ± 3.6	22 ± 3.5	370
Time, min ***						
30	50	0.06	6.6 ± 0.2	9.4 ± 0.3	21 ± 3.4	350
60	42	0.01	10 ± 1.6	15 ± 2.6	32 ± 2.8	570
120	47	0.01	2.3 ± 0.4	2.9 ± 0.5	22 ± 3.3	480

*—at constant molar ratio of RAFT agent:Initiator = 1:10 and irradiation time of 60 min. **—at constant time 60 min and concentration of ST 2 vol.%. ***—at constant concentration of ST 2 vol.% and molar ratio of RAFT agent:initiator = 1:10.

**Table 3 membranes-13-00523-t003:** Elemental composition of PET TeMs surface calculated from EDX spectra.

Sample	Atomic Content, %		
	C	O	S
Pristine PET TeMs	70	30	-
PET TeM-PS Raft agent:Initiator = 1:10 (degree of grafting—2.6 ± 0.02%)	71	29	0.06

**Table 4 membranes-13-00523-t004:** Area under the peak at 700 cm^−1^ calculated from normalized spectra at 1410 cm^−1^ of grafted PET TeMs with styrene at different parameters.

Sample	Area_700_/Area_1410_
Concentration of monomer, vol.% (at constant molar ratio of RAFT agent:Initiator = 1:10 and irradiation time of 60 min)	
1	0.38
2	0.39
4	0.40
RAFT agent:Initiator, molar ratio (at constant time 60 min and concentration of ST 2 vol.%)	
1:2	0.40
1:10	0.39
1:100	0.92
Time, min (at constant concentration of ST 2 vol.% and molar ratio of RAFT agent:initiator = 1:10)	
30	0.30
60	0.39
120	28

**Table 5 membranes-13-00523-t005:** Main parameters of PET TeMs-g-PS before and after grafting with 4-VP.

Reaction Parameters	Grafting Degree, %	Conversion, %	Theoretical Mw of Polymer, ×10^3^ g/mol	Contact Angle at pH, ±3 °			Pore Diameter, ±0.1 μm
				2	7	9	
Pristine PET TeMs	-	-	-	85	67	84	2.0
PET TeMs-g-PS	2.6 ± 0.02	25 ± 2.7	590 ± 61	97	97	97	1.8
Concentration of monomer, vol.% *							
1	0.67 ± 0.03	0.20 ± 0.03	2.7 ± 0.30	65	94	77	1.7
2	1.2 ± 0.02	9.6 ± 0.42	220 ± 9.7	58	97	95	1.7
4	3.3 ± 0.14	5.6 ± 0.53	260 ± 25	49	84	81	1.7
RAFT agent: Initiator, molar ratio **							
1:2	0.68 ± 0,04	5.8 ± 0.49	130 ± 11	78	94	86	1.7
1:10	1.2 ± 0.02	9.6 ± 0.42	220 ± 9.7	58	97	95	1.7
1:100	1.4 ± 0.07	13 ± 0.6	310 ± 14	69	89	78	1.7
Time, min ***							
30	0.5 ± 0.28	9.5 ± 0.13	220 ± 3.1	55	96	76	1.7
45	1.2 ± 0.02	9.6 ± 0.42	220 ± 9.7	58	97	95	1.7
60	0.65 ± 0.02	9.03 ± 0.49	197 ± 12	57	99	73	1.7

*—at constant molar ratio of RAFT agent:initiator = 1:10 and irradiation time of 45 min. **—at constant time 45 min and concentration of 4-VP 2 vol.%. ***—at constant concentration of 4-VP 2 vol.% and molar ratio of RAFT agent:initiator = 1:10.

**Table 6 membranes-13-00523-t006:** Surface properties of PET TeMs at different graft polymerization of 4-VP on PET TeMs-g-PS reaction parameters.

Reaction Parameters	ω, ±0.01 mN/m	γ^p^, ±0.01 mN/m	R_a_, nm	R_q_, nm	F, nN	E, ±200 MPa
Pristine PET TeMs	53	6.7	12 ± 2.5	16 ± 4.5	21 ± 2.5	530
PET TeMs-g-PS	42	0.01	10 ± 1.5	15 ± 2.6	32 ± 2.8	570
Concentration of 4-VP, vol.% *						
1	46	0.05	12 ± 2.1	17 ± 3.3	18 ± 2.7	280
2	48	0.54	11 ± 1.8	15 ± 2.6	22 ± 3.1	290
4	43	2.01	16 ± 3.7	22 ± 5	25 ± 4.1	270
RAFT agent: Initiator, molar ratio **						
1:2	37	0.71	20 ± 2.4	26 ± 3.1	15 ± 2.4	290
1:10	48	0.54	11 ± 1.8	15 ± 2.6	22 ± 3.1	290
1:100	45	0.72	13 ± 1.9	17 ± 2.3	27 ± 6.0	300
Time, min ***						
30	47	0.00	6.6 ± 0.2	9.4 ± 0.35	28 ± 2.2	350
45	48	0.54	11 ± 1.8	15 ± 2.6	22 ± 3.1	290
60	44	0.01	22 ± 3.7	29 ± 4.7	27 ± 8.7	310

*—at constant molar ratio of RAFT agent:initiator = 1:10 and irradiation time of 45 min. **—at constant time 45 min and concentration of 4-VP 2 vol.%. ***—at constant concentration of 4-VP 2 vol.% and molar ratio of RAFT agent:initiator = 1:10.

**Table 7 membranes-13-00523-t007:** Elemental composition of PET TeMs surface calculated from EDX spectra.

Sample	Atomic Content, %			
	C	O	S	N
PET TeMs	70	30	-	-
PET TeMs-g-PS-RAFT (degree of grafting—2.6 ± 0.02%)	71	29	0.06	-
PET TeMs-g-PS-g-P4VP-RAFT (degree of grafting—1.2 ± 0.02%)	67	27	0.05	5.9

**Table 8 membranes-13-00523-t008:** Separation efficiency and droplet sizes before and after separation in the range from 0.1 to 10 µm.

	Direct Emulsion Separation			Reverse Emulsion Separation		
	D, μm before	D, μm after	R, %	D, μm before	D, μm after	R, %
Water–Cetane	0.6	not detected	97–100	4.9	not detected	97–100
Water–Hexane	3.1	not detected	97–100	0.44	not detected	95–100
Water–Cyclohexane	4.5	not detected	97–100	0.71	not detected	95–100

**Table 9 membranes-13-00523-t009:** Comparison of obtained experimental results with results described in the literature.

Membrane	Composition of the Emulsion	Membrane Property in Separation	Flux, L/h·m^2^	Pressure	Separation Efficiency, %	Reference
Cellulose acetate/Nylon 66/Dimethyl Sulfoxide (D1)	Hexane:water	Oil rejection	33	Applied 1.5 × 10^3^ mbar	90	[36]
Cellulose acetate/Nylon 66/Formic Acid	Hexane:water	Oil rejection	23	Applied 1.5 × 10^3^ mbar	70	[36]
Polystyrene@ Fe_3_O_4_ nanofiber membrane	Hexane–water	Superhydrophobicity/superoleophilicity	5000	Without external pressure.	96	[37]
Polyethylene (PP) membrane grafted with poly(2-dimethylaminoethyl methacrylate) at pH9	SDS-stabilized diesel-in-water	High hydrophilicity and oleophobicity	60	Vacuum 1000 mbar	80–100 (by light transmission of the permeate)	[38]
Fluorinated SiO_2_-sprayed PVDF membrane	Water–petroleum ether	Superhydrophobicity	2400	By gravity force (height of ca. 10 cm)	100	[39]
Cu mesh with nanoparticles SiO2 coating	Oil/water	Superhydrophilicity-superoleophobicity	14,000	By gravity	99	[40]
Mesh Cu-3,5-di(trifluoromethyl) phenyl	Oil/water	Superhydrophobicity and superoleophilicity	25,000	By gravity	98	[41]
PET TeMs-trichloro(octyl)silane	Chloroform–water	Hydrophobicity	1100	Vacuum 700 mbar	99	[1]
PET TeMs-trichloro(octyl)silane	Cetane–water	Hydrophobicity	270	Vacuum 700 mbar	100–99	[1]
PET TeMs-stearyl methacrylate	Hexadecane/water	Hydrophobicity	2100	Vacuum 600 mbar	97	[31]
PET TeMs-stearyl methacrylate	Chloroform/water	Hydrophobicity	4000	Vacuum 900 mbar	97	[31]
PET TeMs-PS-P4VP at pH2	Water:cetane	Hydrophilicity	5200	Vacuum 900 mbar	97–100	Present study
PET TeMs-PS-P4VP at pH9	Hexane:water	Hydrophobicity	7400	Vacuum 900 mbar	95–100	Present study

## Data Availability

The data presented in this study are available on request from the corresponding author.

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
