# Peer review of "Stimuli-Responsive Track-Etched Membranes for Separation of Water–Oil Emulsions"

_membranes, 2023, doi:10.3390/membranes13050523_

Round 1

Reviewer 1 Report

The authors developed a pH-responsive membrane with polystyrene-b-poly(4-vinyl pyridine) for water/oil emulsion separation. The fabricated membrane showed hydrophobicity at pH 7-9 and relatively hydrophilicity at pH 2. With the membranes, they separated oil-in-water emulsion and water-in-oil emulsion. However, considering the novelty and overall results, I rejected this paper. The specific reason is as follows.

1. The resolution of some figures is too bad (ex. Figure 4,5,6,10)

2. The readability of tables is poor.

3. In figure 9, it is hard to see the shape of the water droplet.

4. Please check the significant digits.

5. There are insufficient background researches about the oil/water separation and pH-responsive membrane in the introduction section.

6. The pH-reponsive property of polystyrene-b-poly(4-vinyl pyridine) is well known already. What's the novelty of this work?

7. The authors should compare with various oil/water separation membrane in aspect of separation performance. Is the flux same or superior to the previously developed membrane?

8. Why is cetane used among many oils?

9. Why is the pH-responsive membrane needed for oil/water emulsion separation? If the feed solution's pH is not suitable, is it necessary to adjust the pH?

Author Response

We thank the reviewers for their interest in our work and helpful comments that will improve the manuscript and we have tried to do our best to respond to the points raised. As indicated below, we have checked all comments made by the reviewers and have made necessary changes following their remarks.

Response to Reviewer  #1

Comment 1: The resolution of some figures is too bad (ex. Figure 4,5,6,10)

Response: The resolution of Figure 5 was improved.  The insufficiently high resolution of the figures 4 and 10 (AFM images) is caused by an abrupt height displacement of the AFM probe due to adhesion to a dense PS layer. At the same time, we consider that resolution of SEM and AFM images are enough for evaluation of roughness and changes in the pore diameters.

Comment 2: The readability of tables is poor.

Response: We agree that tables 1 and 5 contain a lot of information, however we tried to summarized full information, which is necessary for  and characterization of the samples.

Comment 3: In figure 9, it is hard to see the shape of the water droplet.

Response: Quality of the images have been and now the shape of the drops are clearly visible.

Comment 4: Please check the significant digits.

Response: The significant digits in the manuscript have been checked, and corrections have been made.

Comment 5: There are insufficient background researches about the oil/water separation and pH-responsive membrane in the introduction section.

Response: We have improved the introduction. All corrections are colored in yellow.

Comment 6: The pH-responsive property of polystyrene-b-poly(4-vinyl pyridine) is well known already. What's the novelty of this work?

Response: The novelty of this work is that for the first time the process of UV-initiated grafting RAFT block copolymerization of ST and 4-VP on PET TeMs was studied. The reaction parameters were established at which the change of hydrophilic-hydrophobic properties of the membrane surface was achieved with the change of pH-medium with preservation of the pore structure.  For the first time, track-etched membranes were tested in the separation of both direct and reverse emulsions.

Comment 7: The authors should compare with various oil/water separation membrane in aspect of separation performance. Is the flux same or superior to the previously developed membrane?

Response: Comparison analysis was added to the section 3.3, Table 8.

Comment 8: Why is cetane used among many oils?

Response:  Cetane is most common used oils and there is a lot of information on cetane in the article. However, we expanded section 3.3 and revised manuscript presents the results of testing direct and inverse emulsions for cetane-water, hexane-water, cyclohexane-water, to show effectiveness for different kinds of solvents.

Comment 9: Why is the pH-responsive membrane needed for oil/water emulsion separation? If the feed solution's pH is not suitable, is it necessary to adjust the pH?

Response: Preparation of membranes with switchable surface properties is an important task, which is the subject of many works (for instance, they are summarized in https://doi.org/10.1016/j.progpolymsci.2018.06.009) and they can be used for controllable oil-water separation (as smart separation material) as well as they also can find application in chemical processes and sensorics.  For this type of membranes, it is necessary to adjust pH.

Author Response

We thank the reviewers for their interest in our work and helpful comments that will improve the manuscript and we have tried to do our best to respond to the points raised. As indicated below, we have checked all comments made by the reviewers and have made necessary changes following their remarks.

Response to Reviewer  #2

Comment 1: In the introduction, please give brief explanation of superhydrophilic, underwater superoleophobic, superhydrophilic/superoleophobic surfaces for oil-water separation. Also, mention what two modes of wetting the pH-sensitive separation membranes switches between both for previous studies and this study. 

Response: Introduction was expanded according to the comment (please see lines 40-46, 56-60).

Comment 2: What is the photosensitization in line 90? Please include brief explanation.

Response: Photosensitization is the process of photooxidation of radiolysis products after irradiation with heavy ions, which allows increasing chemical etching rate of ion-tracks. The description of the photosensitization process was also added to the manuscript (experimental part, section 2.2).

Comment 3: It would be better to give oil(cetane) contact angle data also

Response: Also, it should be noted that obtained pH-responsive PET TeMs, after soaking at pH9 (membrane become hydrophobic), instantly allow pure cetane, hexane, and cyclohexane to pass through the pores of the membranes. However, after soaking the membrane at pH2, CA of cetane, hexane, and cyclohexane is 29, 28, and 26°, respectively, and during filtration testing, the oil remains on the surface of the membrane and does not leak out.

Comment 4: What is the droplet size of direct or reverse emulsion. Did you measured DLS or look through microscope. After the water/oil emulsion separation test, did you analyze the purity of filtrate or residue on the filter. For example, oil content in water or water content in oil.

Response: Emulsions before and after separation were analyzed using DLS. For instances, the droplet size for the direct emulsions was 0.6, 3.1, and 4.5 μm for cetane, hexane, and cyclohexane as the organic part, respectively. The droplet size of the reverse emulsions was 4.9 μm. After emulsion separation, DLS analysis did not detect an emulsion even after dispergation of the filtrate by IKA T18 digital Ultra-Turrax disperser. Thus, only trace amount of oil in water can be. Trace amount of oil in water or water in oil was not studied. However, we will consider your comment for future research.

Comment 5: As the concentration or UV irradiation time of styrene increased, roughness decreased (Table 2). In contrast, roughness increased for 4-VP as these parameters increased (Table 6). Why do you think so.

Response: Grafting of PS with increasing concentration to 1, 2 and 4 vol.% smooths the membrane surface, thereby decreasing the mean Ra values (from 11.6±2.5 to 14.0±3.1, 10.0±1.5 and 2.2±0.4 respectively) and Rq mean Roughness (from 16.0±4.5 to 20.5±5.0, 15.3±2.6 and 2.8±0.5 respectively) of membrane surface roughness. Further grafting of P4VP to the PS ends on the membrane surface with increasing concentrations to 1, 2 and 4%, on the contrary, leads to an increase in Ra values (from 10.0±1.5 to 12.14±2.15, 11.37±1.81 and 16.12±3.69, respectively) and Rq (15.28±2.6 to 16.64±3.31, 15.22±2.61 and 21.72±5.01, respectively) of membrane surface roughness compared to PET TeMs-PS. Referring to the degree of grafting, conversion and theoretical polymer mass, the dependence of the change in roughness of the modified membranes can be explained as follow. As can be seen from Tables 2 and 6, these parameters increase as the monomer concentration increases, but in varying degrees of increase depending on the monomer. That is, grafting of P4VP with increasing concentration to 1, 2 and 4 % increases the graft degree to 0.67±0.03, 1.19±0.02 and 3.29±0.14 %, conversion to 0.20±0.03, 9.63±0.42 and 5.56±0.53 % and theoretical molecular weight to 2.72±0.3, 223.40±9.74 and 258.09±24 66 ×103 g/mol, which is about 3-4 times lower than for styrene grafting: graft degree to 0.21±0.10, 2.61±0.02 and 8.65±1.03%, conversion to 17.29±0.30, 24.85±2.72 and 35.84±0.89% and theoretical molecular weight to 192.35±3.35, 589.26±61.04 and 1604.09±39.83 ×103 g/mol. Thus, the grafted dense PS layer covers a larger membrane surface area than that of P4VP, due to which irregularities are formed after grafting of P4VP and the roughness on the membrane surface increases.

Comment 6: In Table 5, WCA’s are lower at pH 9 than at pH 7 for many conditions. Please give your explanation.

Response: It is assumed that CA at pH9 is lower that at pH 7 since RAFT agent at the end of the grafted polymer can contain carboxyl group which can be ionized at pH9 and also influence on the pH value.  This is also confirmed by the fact that after the destruction of the RAFT agent (by heating), an increase in the CA is observed at pH 9.

Comment 7: In Figure 9, please show the picture of water droplet rather than OCA calculation picture. I cannot see the shape of water droplet clearly.

Response: The quality of the images has been improved by brightening the background and now the borders of the drop are clearly visible.

Comment 8: Line 379 : Figure 1a → Figure 8a

Response: It was corrected.

Comment 9: As the authors mentioned that P4VP could destruct in UV irradiation. Do you have future plan for this? It would be better to include in conclusion.

Response: P4VP as well as PET and many others polymer degrade under UV-irradiation (10.1186/2193-1801-2-398). Degradation rate depends of UV-power, solvent, and etc. In our study we just recorded that with increasing time of irradiation grafting degree (measured by weighting) decreased, we attributed these observations to the fact that the weight of the membrane decreased due to the degradation of the polymer (this can be either PET substrate, grafted PS or grafted P4VP). So, we decided not increase time of irradiation.

Reviewer 3 Report

Comments:

1.      The authors reported that PET TeMs were obtained with pore diameters of 2.01±0.1 μm, how did the authors get the pore size of 2.01μm and the error is 0.1 μm?

2.      What is the novelty of this work? Why did a membrane with emulsion separation need pH responsive?

3.      How to verify and calculate the grafting degree in Fig. 3?

4.      The resolution of Fig. 4d is not high enough.

5.      The elements distribution in Fig. 5 is difficult to read.

6.      What is the porosity of these as-prepared membranes? What are the influence factors?

7.      The manuscript took too much space on discussing the preparation of the membrane instead of testing the separation efficiency and explain the mechanism.

Author Response

We thank the reviewers for their interest in our work and helpful comments that will improve the manuscript and we have tried to do our best to respond to the points raised. As indicated below, we have checked all comments made by the reviewers and have made necessary changes following their remarks.

Response to Reviewer  #3 

Comment 1: The authors reported that PET TeMs were obtained with pore diameters of 2.01±0.1 μm, how did the authors get the pore size of 2.01μm and the error is 0.1 μm?

Response: We agree with the reviewer’s comment. The average pore diameter were calculated based on SEM images. Values of average pore diameters were corrected.

Comment 2: What is the novelty of this work? Why did a membrane with emulsion separation need pH responsive?

Response: The novelty of this work is that for the first time the process of UV-initiated grafting RAFT block copolymerization of ST and 4-VP on PET TeMs was studied. The reaction parameters were established at which the change of hydrophilic-hydrophobic properties of the membrane surface was achieved with the change of pH-medium with preservation of the pore structure.  For the first time, track-etched membranes were tested in the separation of both direct and reverse emulsions.

Comment 3: How to verify and calculate the grafting degree in Fig. 3?

Response: The degree of grafting was determined according to the equation (1) by the difference in membrane weight before and after the grafting as shown in previously published works [1,22].

                              (1)

where  – the degree of grafting,  – the weight of the membrane before grafting, the weight of the membrane after grafting. To remove electrostatic charge, antistatic ionizer was used before weighing.

Comment 4: The resolution of Fig. 4d is not high enough.

Response: The insufficiently high resolution of the 4d image is caused by an abrupt displacement of the AFM probe due to adhesion to a dense grafted PS layer with high grafting degree.

Comment 5: The elements distribution in Fig. 5 is difficult to read.

Response: The images in Figure 5 have been improved.

Comment 6: What is the porosity of these as-prepared membranes? What are the influence factors?

Response: The pristine PET TeMs has porosity of 5 %. Grafting of the block copolymer leads to a decrease in porosity to 4%, resulting from a decrease in the pore diameter due to polymer grafting.  It is well known that porosity significantly affects the performance of membranes. But performance is also affected by membrane thickness, tortuosity of channels, pore size. Track-etched membranes are distinguished by the fact that they have a small thickness (12-24 mm), non-tortuosity of the channels, we also used channels with large diameters, which made it possible to increase the performance of TeMs. And finally, the fluxes of obtained membranes superior in many cases to nanofiber and other commercial membranes.

Comment 7: The manuscript took too much space on discussing the preparation of the membrane instead of testing the separation efficiency and explain the mechanism.

Response: We have improved the section 3.3 Testing of pH- responsive membranes in water-oil emulsion separation.

Round 2

Reviewer 1 Report

There are still low-resolution images in manuscripts. There are also many typos in manuscripts. The author should check the units, such as L/h*m2 or L/min*m2, and if the values are accurate. The author compared the separation performance with the literature, but I think the applied or vacuum pressure for filtration was not considered.

Author Response

We thank the reviewer for their interest in our work and helpful comments that will improve the manuscript and we have tried to do our best to respond to the points raised. As indicated below, we have checked all comments made by the reviewer and have made necessary changes following their remarks.

Response to Reviewer #1

Comment 1: There are still low-resolution images in manuscripts.

Response: We have made every effort to improve the quality of the images in the manuscript, but, due to the limitations of the AFM and SEM instrument, we were unable to achieve high resolution images. Nevertheless, we still believe that the images presented are valuable for our work and sufficient to understand the results of the study of the roughness and pore diameter of the membrane surface.

Comment 2: There are also many typos in manuscripts.

Response: Typos have been corrected.

Comment 3: The author should check the units, such as L/h*m2 or L/min*m2, and if the values are accurate.

Response: The units have been checked and corrected. In particular, the flow unit has been changed from L/min*m2 to L/h*m2.

Comment 4: The author compared the separation performance with the literature, but I think the applied or vacuum pressure for filtration was not considered.

Response: When comparing the separation fluxes with the literature data, the pressure was taken into consideration. Changes have been added to Table 8.

Reviewer 2 Report

The manuscript was successfully revised according to comments

Author Response

We thank the reviewer for their interest in our work and their helpful comments, which improved the manuscript